# Neural Network Models for Driving Control of Indoor Autonomous Vehicles in Mobile Edge Computing

**DOI:** 10.3390/s23052575

**Published:** 2023-02-25

**Authors:** Yonghun Kwon, Woojae Kim, Inbum Jung

**Affiliations:** Department of Computer Information and Communication Engineering, Kangwon National University, Chuncheon 24341, Gangwondo, Republic of Korea

**Keywords:** neural network model, indoor autonomous driving, LiDAR sensor, resource usage, mobile edge computing

## Abstract

Mobile edge computing has been proposed as a solution for solving the latency problem of traditional cloud computing. In particular, mobile edge computing is needed in areas such as autonomous driving, which requires large amounts of data to be processed without latency for safety. Indoor autonomous driving is attracting attention as one of the mobile edge computing services. Furthermore, it relies on its sensors for location recognition because indoor autonomous driving cannot use a GPS device, as is the case with outdoor driving. However, while the autonomous vehicle is being driven, the real-time processing of external events and the correction of errors are required for safety. Furthermore, an efficient autonomous driving system is required because it is a mobile environment with resource constraints. This study proposes neural network models as a machine-learning method for autonomous driving in an indoor environment. The neural network model predicts the most appropriate driving command for the current location based on the range data measured with the LiDAR sensor. We designed six neural network models to be evaluated according to the number of input data points. In addition, we made an autonomous vehicle based on the Raspberry Pi for driving and learning and an indoor circular driving track for collecting data and performance evaluation. Finally, we evaluated six neural network models in terms of confusion matrix, response time, battery consumption, and driving command accuracy. In addition, when neural network learning was applied, the effect of the number of inputs was confirmed in the usage of resources. The result will influence the choice of an appropriate neural network model for an indoor autonomous vehicle.

## 1. Introduction

Cloud computing is based on servers with infinite resources and computational power, which provide high-quality services to many mobile devices that have limited computational resources. However, when many mobile devices request their computing simultaneously, the severe data traffic delays the network bandwidth, making it difficult to cope with such urgent requests. Mobile edge computing has emerged as a possible solution to the problems that occur in a cloud computing environment. Mobile edge computing does not transmit the processing of urgent requests to the cloud server but instead proceeds to edge devices deployed in the region where the requests occur. Additionally, it provides services, such as vehicles driven by users, transportation means, and remotely controlled drones, by implementing various platforms [1,2]. Autonomous driving is known as a representative service case of mobile edge computing. For example, in the past few years, due to COVID-19, some cases have been studied for medical staff’s non-contact service [3,4]. Urgent requests and driving information that occurs during autonomous driving are not sent to the cloud; however, the self-moving vehicle handles them on its own, enabling rapid response [5].

Unlike outdoor autonomous driving, indoor autonomous driving has restrictions and cannot employ a location recognition method using a Global Positioning System (GPS). Sensors necessary for indoor driving and various techniques based on PLAN (positioning, localization, and navigation) technology have been proposed to address this problem. Sensors used in autonomous driving include cameras, LiDAR (light detection and ranging), inertial sensors, and ultrasonic waves. Furthermore, as indoor autonomous driving research, the SLAM (Simultaneous Localization and Mapping) uses various sensors to identify the interior structure of the room and presents a method for performing map creation and location recognition simultaneously. In addition, studies on vehicle control with machine learning algorithms, such as algorithms for calculating driving paths and object recognition, are being conducted [6]. Indoor autonomous driving is being studied on various hardware platforms. However, studies on small indoor autonomous vehicles using price-competitive hardware platforms such as the Raspberry Pi have recently been proposed [7]. Moreover, devices such as the Raspberry Pi have lower specifications in regard to CPU computation, memory storage, wireless network safety, and battery performance. The evaluation of resources required for autonomous driving and the accuracy of inferring driving commands are necessary to improve driving time and stability in this operating environment.

In this study, we built an autonomous driving vehicle using the Raspberry Pi and a LiDAR sensor in an indoor environment and evaluated its performance. The driving command of the autonomous vehicle was decided based on the data obtained from the surroundings of the moving vehicle using the LiDAR sensor as input data for neural network learning. The LiDAR sensor calculates the distance based on the time required for light to emit and return after hitting an object. However, the distance to the object may not be accurately measured depending on the distance to the object and the material of the object’s surface when light hits the object [8]. To solve this problem, when the noise occurs in the data measured from the LiDAR sensor, it is removed using the data around the noise and then used for neural network learning. Furthermore, the autonomous vehicle predicts one of the five driving commands appropriate for the current location according to the learning of the neural network. The vehicle performs autonomous driving according to the selected driving command.

The amount of data measured with the LiDAR sensor is adjusted based on the nature of the Raspberry Pi with limited resources to ensure that the processing burden can be controlled. In this study, the resource consumption in the neural network learning model was measured by changing the amount of data measured from the LiDAR sensor and evaluating the accuracy of the predicted driving command. If the number of range data input to the neural network increases, the accuracy of the predicted driving command increases, while the resources used for calculation also increase. Furthermore, we evaluate the performance of the neural network model according to the number of range data points from the LiDAR sensor used. Six neural network models are created according to the number of LiDAR sensor data points in the indoor autonomous driving circular track. The requirements for indoor autonomous driving have conflicting aspects. We evaluate the performance of each learning model’s learning speed, response time, battery resource consumption, and prediction accuracy in six neural network models. Performance evaluation results reveal a trade-off between the permanence and safety of indoor autonomous vehicles operating in an edge environment. Based on these results, it is possible to obtain a resource consumption method suitable for the given resource conditions and current autonomous driving vehicle mission.

The contributions of this paper are as follows:Proposing an autonomous driving model for mobile edge computing.Designing a neural network model using only LiDAR sensor.Implementing an indoor vehicle based on Raspberry Pi and LiDAR.Performance evaluation of inference models divided by input size and observation of the trade-off.

The remaining sections of the study include the following sections: Section 2 describes the contents of existing studies on indoor autonomous driving. Section 3 describes a neural network model for indoor autonomous driving. Section 4 describes the implemented autonomous vehicle and the experimental environment. Section 5 presents how the performance is evaluated according to the established criteria, the efficient use of resources, and the trade-off point between requirements. Section 6 discusses the conclusions and future work of this study.

## 2. Related Work

### 2.1. Mobile Edge Computing

The edge exists between the cloud server and mobile device and is physically close to the user. The data generated by the device at the end of the network is transmitted to the mobile edge for processing, and the user is provided with the result. Furthermore, it is possible to alleviate the time delay problem caused by receiving the result after transmission to the cloud server because of this process. Additionally, the latency reduction effect of mobile edge computing is thus beneficial in the field of autonomous driving. Response time is directly related to safety in the domain of autonomous driving. Sensors such as LiDAR, cameras, and ultrasonic waves are used to observe the surrounding environment and generate data in autonomous driving. Complex computations, such as obstacle and environment recognition and map generation, are performed based on the generated data. In such an environment, delays in sending the data to the cloud server and waiting for the results can lead to severe accidents, depending on the results. Therefore, mobile edge computing is helpful in fields such as autonomous vehicles, requiring real-time data operations guarantees [9].

Mobile edge computing is in resource constraints. Unlike servers, mobile edge devices are small and have a limited power supply. Therefore, there is a lack of computing power. In the mobile environment, connection to the upper edge server or cloud may not be guaranteed, depending on the environment. As a result, a mobile edge should be able to perform tasks in on-device environments.

### 2.2. Indoor Autonomous Driving

Autonomous driving in an indoor environment differs from driving in an outdoor environment in terms of detection and recognition technologies. In an outdoor environment, relatively accurate location recognition is achieved through a Global Positioning System (GPS), and location information can be obtained from road traffic information and topographical features. However, it is difficult to use the GPS indoors because the satellite signal reception is not smooth, the location recognition accuracy is lowered, and the number of floors cannot be determined in the case of a multi-story building [6,10]. Therefore, it is important to recognize the location using the installed sensor data to perform indoor autonomous driving.

Indoor autonomous driving is being studied in various ways. SLAM is a technique that simultaneously performs map creation and location recognition. Depending on the method, it can be divided into using a visual sensor such as a camera and a method using a LIDAR sensor [11,12,13,14]. LiDAR-based SLAM can be effectively used in cases where camera sensors cannot be utilized, such as in underground mines [15]. Machine-learning methods are also used to perform indoor autonomous driving. In [16], corrected the wheel’s drift during driving using a backpropagation neural network and CNN. In other studies, there was a case of implementing a neural network model based on a vision sensor and LiDAR data [17,18].

Raspberry Pi is widely used as an autonomous driving platform because Raspberry Pi is inexpensive. It lacks performance compared to Nvidia’s Jetson TX2, which is used in many studies [7]. SLAM is operated in [19,20], but the map is small, or the vehicle control model is not presented. Refs. [21,22] implemented a vehicle control model using a camera and lidar sensor, but map production and location recognition were not performed. In this resource-constrained environment, a lightweight and fast model is needed among mapping and vehicle control models.

### 2.3. Backpropagation Neural Network

The neural network model used in this study is based on backpropagation. Backpropagation neural networks are algorithms that adjust the connection strength by propagating the errors from the output layer of the hierarchical neural network to the input layer [23]. Figure 1 shows the structure of the backpropagation neural network. A neural network consists of an input layer, several hidden layers, and output layers. Each node is fully connected, meaning each layer’s nodes are connected to all the nodes of the next layer. The initial value of the input layer in a backpropagation neural network passes through the intermediate layer composed of several layers, and finally, the output layer outputs the value. Furthermore, learning is conducted so that the connection strength between layers is adjusted inversely by comparing the output value with the previously provided correct answer value if an error is generated. Additionally, the learning is terminated if the error between the output and the correct answer becomes smaller than the allowable error during the learning process. A neural network model can be constructed based on the connection strength generated after training is completed [24]. A neural network model can be used as a classifier with many features and categories, such as LiDAR data because neural network learning is an end-to-end learning that can easily modify input and output size. For example, a linear classifier, SVM (Support Vector Machine), is limited to two categories. Additionally, rule-based classifiers or decision tree models are unsuitable for tasks with many features, such as LiDAR data, because users must specify constraints or rules for each feature [25].

## 3. Neural Network Models on Autonomous Driving

Figure 2 is a context diagram of the indoor autonomous driving system proposed in this paper. The autonomous driving vehicle uses the range data measured from the LiDAR sensor to determine a suitable driving command for the current state of the vehicle and moves by itself. As shown in the figure, the proposed system consists of three modules: Sensing, Prediction, and Driving. The Sensing module measures and processes LiDAR data and delivers it to the Prediction module. LiDAR data are measured through the middleware of the robot operating system (ROS), which allows data from various sensors to be used. ROS middleware supports the use of various sensors. The Prediction module uses a neural network to predict the driving commands that the autonomous driving vehicle should perform.

The training of the neural network is performed based on data measured in advance from the path the autonomous vehicle travels. The driving command generated as a result of learning is delivered to the Driving module, which controls the vehicle based on these commands. The driving command is transmitted to the motor to operate the wheels of the autonomous driving vehicle. Each command is carried out by adjusting the wheel’s speed and direction of rotation.

Algorithm 1 is a pseudo-code that contains the overall process for the neural network model. It repeats according to the sensing cycle of LiDAR. Lines 12 to 18 perform denoising and normalization on range data measured from LiDAR. If the raw data value is 0, the denoising model interpolates the value by referring to the data around the sensing point. In line 18, normalization is performed based on the largest value among the lidar range values measured so far. The normalized values constitute N, the input data for the prediction model. In line 20, the prediction proceeds. The vehicle drives according to the command inferred from the 21.
**Algorithm 1 Driving Command Prediction**1.***R*****: raw range data from LiDAR Sensor**2.*Command*: driving command3.*N*: normalized range data4.*m*: number of input size5.*max*(*R*): largest range data in R 6.*predict*(*x*): driving command prediction model7.*denoise*(*x*): denoising model8.*drive*(*x*): driving model9.{*r_1_, r_2_, …, r_m-1_, r_m_*}∈R10.{*n_1_, n_2_, …, n_m-1_, n_m_*}∈N11.**repeat**12.  **for** *i* ←1 to *m* **do**13.   **if**
*r_i_* = 0 **then**14.    *r_i_* ← (*denoise*(*r_i-_*_1_)+ *denoise* (*r_i+_*_1_))/215.
   **else**
16.    *r_i_* ← *denoise* (*r_i_*)17.
   **end**
18.   *n_i_*← *r_i_* / *max*(*R*)19.
  **end**
20.
  *Command*
*←*
*predict(N)*
21.  *drive*(*Command*)22.**end**

### 3.1. Learning Data Configuration

In this study, an indoor autonomous driving environment was established, and data were measured using a LiDAR sensor. The LiDAR sensor rotated 360° and produced 360 range data in a single measurement. The learning data for autonomous driving were based on the range data measured with the LiDAR sensor. The LiDAR sensor constituting the autonomous moving body rotated 360° counterclockwise with reference to the front and generated 360 range data every 0.2 s.

Furthermore, six learning models were created for performance evaluation according to the input size of the neural network. The measurement width of the data was adjusted and classified into six sets of range data for learning each model. Figure 3 shows the distribution LiDAR data. Equations (1)–(6) express the LiDAR dataset used in this study. Each dataset is represented by Data_n_, where n denotes the number of range data points constituting one dataset. Furthermore, each data point is distributed at regular intervals at 360°.

Equation (1) is the range data in eight directions. Each datum is distributed at 45° intervals, with a total of eight data points. Equation (2) was measured at 20° intervals, and the number of data is 18. Equation (3) represents 45 range data measured at 8° intervals, and Equation (4) represents 90 range data measured at 4° intervals. Equation (5) is a dataset of 180 ranges measured at 2° intervals. Finally, Equation (6) is 360 intact range data without increasing the interval.
*Data*_8_ = [*range*_0_, *range*_45_, . . . . . , *range*_270_, *range*_315_](1)
*Data*_18_ = [*range*_0_, *range*_20_, . . . . . , *range*_320_, *range*_340_](2)
*Data*_45_ = [*range*_0_, *range*_8_, . . . . . , *range*_344_, *range*_352_](3)
*Data*_90_ = [*range*_0_, *range*_4_, . . . . . , *range*_352_, *range*_356_](4)
*Data*_180_ = [*range*_0_, *range*_2_, . . . . . , *range*_356_, *range*_358_](5)
*Data*_360_ = [*range*_0_, *range*_1_, . . . . . , *range*_358_, *range*_359_](6)

Figure 4 depicts an example of driving commands that must be executed according to the location when driving in an indoor driving environment. There are five driving commands: ①—Turn Left, ②—Accelerate Left, ③—Go Straight, ④—Accelerate Right, and ⑤—Turn Right. The order is divided into situations where you have to go straight and those where you have to turn. In a straight environment, the center of the vehicle body is moved to the center of the driving path for stable driving without colliding with walls on the left and right or obstacles. The situation of driving straight is divided into three commands, ②, ③, and ④. Furthermore, the ③—Go Straight command is executed when the vehicle is located at the center of the passage. The ②— Accelerate Left command is executed when the vehicle body is biased to the left rather than the center. Conversely, if the vehicle body is biased to the right rather than the center, the ④— Accelerate Right command is executed. A junction is a point at which a vehicle can move in all directions according to a selected driving command. Additionally, the corner section determines the direction in which the vehicle can move. ①—Turn Left and ⑤—Turn Right commands are executed at junction points or corner sections. A driving command is executed along a path that can be driven in the corner section; however, the driving command may differ even at the same junction point.

### 3.2. Noise Removal

The LiDAR sensor used in this study generates noise because it does not return normal values if the range is not measured. The learning result may vary depending on the data sampling interval even at the same point, and the accuracy of the trained neural network model may be reduced if noise is present. Thus, we improved the reliability of learning by removing noise.


(7)
$$f\left(x_{n}\right)=\left\{\begin{array}{c} x_{n}\;\;\;\;\;\;,\;\left(x_{n}\neq 0\right)\\ \frac{f\left(x_{n-1}\right)+f\left(x_{n+1}\right)}{2},\left(x_{n}=0\right) \end{array}\;\left(n=1,2,\ldots ,360\right)\right.$$


Equation (7) performs the function of filtering noise. If the range data are normally measured, the value is stored; if the value is 0, it is regarded as noise, and the value is predicted using adjacent data. Figure 5 shows the distribution of range data measured at a specific point. The x-axis of the graph represents the order of 360 data points from 0° to 359°, the point at which the data were measured. The y-axis of the graph represents the size of the normalized range data. Figure 6 shows the distribution of data with this noise removed. As mentioned previously, when noise is generated in the data, new data are created based on surrounding data to remove noise.

### 3.3. Neural Network Model

Figure 7 depicts the neural network model used in this study. The model determines the optimal driving command for the current situation based on the input LiDAR range data. The neural network model is in the form of a fully connected layer consisting of an input layer, a hidden layer, and an output layer. The output layer is classified into five driving commands. ReLU was used for the activation function of the middle layer, and Softmax was used for classifying driving commands in the output layer. The loss function of the model was cross-entropy. *n* in Figure 5 represents the size of the input. Furthermore, the size of the input corresponds to the six training datasets classified in Section 3.1. For example, the neural network learning model using Equation (1) data is “Model-8”. Furthermore, the model using the training data of Equation (2) is “Model-18.” “Model-45,” “Model-90,” “Model-180,” and “Model-360” correspond to Equations (3)–(6), respectively. The epochs of the neural network were set to 10,000 for performance evaluation in a limited environment. The learning rate of the model was derived by repeating the learning process. Furthermore, learning was most accurately completed in the limited epoch when the learning rate was set to 0.3.

## 4. Implementation

### 4.1. Indoor Autonomous Driving Vehicle

Figure 8 shows the implemented autonomous vehicle, which is based on the Raspberry Pi 4 Model B. The Raspberry Pi is a small, inexpensive board developed for educational purposes. It has excellent performance compared to the price, and it is easy to develop a system for autonomous driving using a Linux-based OS. The LiDAR sensor that measures data is the SLAMTEC RPLIDAR A1M8 model. The LiDAR sensor rotates 360° and can measure distances from 0.15 to 12 m. The wheels are connected to four 12 V DC motors, and an 11.1 V battery pack is installed to supply power to the main board and the motor driver to control the motors for driving. Table 1 shows the specifications of the Raspberry Pi and LiDAR sensor [26,27].

### 4.2. Indoor Autonomous Driving and Data Collection

The autonomous vehicle implemented in 4.1 was used to collect learning data. Figure 9 shows an indoor driving track built for autonomous driving. We placed an autonomous vehicle equipped with LiDAR on an autonomous track and measured the range data between the vehicle and the wall while moving along the track.

Figure 10 shows the data collection path in the circulation track based on the implementation track in Figure 9. The autonomous vehicle started from ‘Start,’ drove in order from ① to ⑦, and stopped at ‘Finish’. The LiDAR data collected during the driving process was stored, and the driving commands required at each location were labeled on the data collected at the corresponding location. Furthermore, 1100 datasets consisting of 360 range data were collected during the collection process. A total of 770 of the collected data were used for learning, and 330 were used for evaluation.

### 4.3. Learning Data Creation

In this study, we evaluated the performance of neural network models classified according to the size of the neural network input. Furthermore, LiDAR data were measured at 1° intervals of 360° and expressed as a total of 360 data. In addition, the data measurement interval was adjusted to control the number of input data points, and six datasets were created according to the size of the input data. Table 2 lists some of the training data generated for a model with an input size of 360. R_0_ represents the range data measured at 0° from the LiDAR sensor. The range data were normalized to a value between 0 and 1 based on the largest value among the measured data. This is a dataset for learning a classified model. Therefore, the driving command appropriate for the point where the dataset was measured was labeled as the correct answer, as shown in C_0_–C_4_ in the range dataset. The driving commands correspond to “Go Straight,” “Turn Left,” “Turn Light,” “Accelerate Left,” and “Accelerate Right” from C_0_ to C_4_, respectively.

## 5. Performance Evaluation

In this experiment, resource usage was measured according to the structure of the learning model when neural network learning was used in an edge device environment with limited resources, such as the Raspberry Pi. Furthermore, we found a trade-off point between conflicting requirements in the edge computing environment through performance evaluation. Therefore, we discuss the optimal autonomous driving method under the given conditions. In this study, six neural network learning models were created according to the change in the size of the data measured from the LiDAR sensor. Additionally, the confusion matrix, learning rate, response time, battery consumption, and driving command accuracy were used for performance evaluation.

### 5.1. Confusion Matrix

Table 3 shows the results of comparing the performance evaluation of six neural network learning models using a confusion matrix [28]. Despite the same learning model, model performances are different each time by initial network weights. Each of the six models was trained ten times, and we compared performance using the average of ten results. Each index was calculated using Equations (8)–(11).
(8)$$Accuracy=\frac{TP+TN}{TP+FP+TN+FN}$$
(9)$$Precision=\frac{TP}{TP+FP}$$
(10)$$Recall=\frac{TP}{TP+FN}$$
(11)$$F1\;\mathrm{Score}=2\times \frac{Precision\;\times \;Recall}{Precision+Recall}$$

True positive (TP) is when a correct answer is predicted to be the correct answer for all test data. True negative (TN) is when an incorrect answer is predicted to be an incorrect answer. In addition, false positive (FP) is when an incorrect answer is predicted as a correct answer, and false negative (FN) is when a correct answer is predicted as an incorrect answer. Each value is calculated as the average of five driving commands. Accuracy represents the ratio of correct to incorrect predictions for all given results, and it can be seen that it increases in proportion to the input size of the model. When the size of the input increases, it increases by 0.0141 on average. The largest increase occurred between Model-180 and Model-360, where the difference in the size of the input was the largest and the accuracy increased by 0.0219. Precision represents the proportion of actual correct answers compared to predicted correct answers. The recall represents the proportion of predicted correct answers for actual correct answers. In this experiment, the recall was higher than the precision for all models. The F1 Score is the harmonic mean of precision and recall and is an indicator that can more accurately compare performance in an unbalanced data distribution.

In the experiment above, there are many rotation sections in the driving track; therefore, the data to predict the straight-ahead command and the data to predict the rotation command are distributed in a similar ratio. However, there may be many sections in which a direct command is executed depending on the driving environment, and there may be an extremely maze-like environment in which only turn commands are executed. When comparing the F1 score, Model-8 shows a low performance of less than 0.5. Additionally, the middle-level models have values between 0.5 and 0.8, and Model-360, which has the largest input size, shows a value of 0.8275. There is a performance difference of 7.9% in accuracy and a performance difference of 68.05% in F1 Score when comparing Model-8 and Model-360.

### 5.2. Learning Rate

In a resource-limited environment such as a Raspberry Pi, neural network learning is sped up, thereby reducing power consumption during training. However, if the number of data inputs to the neural network is reduced, the number of operations required to learn all parameters once is reduced; however, the number of parameters to be learned is also reduced, resulting in lower accuracy. In addition, increasing the input data may increase the accuracy, but increase the training time. To evaluate this, learning was conducted 10 times for each model until the accuracy reached 0.95 for the six models, and the average of the time required at this time was compared. In the learning process, the maximum number of epochs was set to 500,000.

Figure 11 shows the time taken for each model to reach an accuracy of 0.95. Among the six learning models, Model-8 did not reach 0.95 accuracy until reaching the maximum epoch in the process of learning 10 times. Model-18 took 514.05 s and Model-45 took 306.19 s, and the average epochs of the two models were 49,856 and 29,696, respectively. The number of input data increased by 150% and the learning time decreased by 67.9% when the two models were compared. Model-90 achieved an accuracy of 0.95 over an average of 18,601 epochs in 210.01 s.

In Model-180, the learning time decreased by 71.6% to 122.35 s, and as the number of input data increased, the learning time decreased at the largest rate. Furthermore, the epoch was 9305, which was similar to the result evaluated in the confusion matrix of Section 5.1. Model-360, using all the range data generated by the LiDAR sensor, reached 0.95 when the value of the epoch was 7339 and the learning time was 117.13 s. Furthermore, the learning time was reduced by 29.5% on average whenever the number of range data points to be learned increased.

### 5.3. Response Time

When an autonomous vehicle moves and interacts with surrounding objects in real space, physical collisions and malfunctions may occur in an unexpected situation if the response time is long. In this experiment, the response time was measured as the time taken to predict the driving command because of the discovery of surrounding objects while the autonomous vehicle was moving. Figure 12 shows the response times of the six neural network models. Data were entered 1,000,000 times for each model and compared using the average value, excluding the top 5% and bottom 5% of the response time. The response time increased as the number of inputs to the model increased. The response time of Model-8 was the fastest at 378.903 μs, and the response time of Model-360 was the slowest at 417.985 μs. Furthermore, the response time increased by 2% on average as the input size increased. As the size of the input increased, the number of calculations processed in the input layer of the neural network model more than doubled, resulting in a larger increase in response time.

### 5.4. Battery Usage

Battery resource usage is a factor that greatly affects the activity radius and service time of autonomous driving vehicles because the mobile edge computing environment uses batteries as a basic operating condition. It is necessary to apply a resource-efficient model to autonomous driving vehicles to extend the limited battery life as much as possible. Figure 13 measures battery consumption while periodically inputting LiDAR data into six neural network models and predicting driving commands for 10 min. Each model was measured 10 times, and the averages were compared. Power in the standby state (Idle) was consumed by an average of 80.6 mAh. Model-8, with the least amount of input data to process, consumed 93.2 mAh on average, and Model-360, with the largest amount of input data, consumed 95.8 mAh on average. This experimental result shows that as the number of input data increases, the battery consumption increases, but not significantly in proportion to the number of data. When comparing the standby state and Model-8, the battery consumption increased by 12.6 mAh. As the size of the input increased from Model-8 to Model-360, the battery consumption increased by an average of 0.52 mAh. Furthermore, batteries are consumed to operate multiple hardware components including motors as well as to determine driving commands for an autonomous vehicle to move. These experimental results show that the battery consumption required to calculate the neural network algorithm does not account for a large portion of the total battery consumption of an autonomous vehicle.

### 5.5. Driving Command Accuracy

In this study, we designed a model that predicts LiDAR sensor data as a driving command by using a neural network. As described in Section 5.1, already-prepared evaluation data were used in the confusion matrix to evaluate the prediction accuracy. However, instead of using the prepared evaluation data, the accuracy had to be evaluated using real-time LiDAR sensor data acquired during driving of the autonomous vehicle. In this experiment, the predicted results were compared with actual driving commands (ground truth), when range data were entered into each model.

Figure 14 depicts the experimental results for the six models. The driving route is shown in Figure 10. A total of 153 driving commands were predicted while driving. The driving command to be executed during actual driving was compared with the driving command predicted by the neural network. The movement of the plot lines in the vertical direction indicates the execution of a command while the vehicles move along a straight line.

Figure 15 presents a comparison of the real-time driving command prediction accuracy of the six models. Model-360 showed the highest accuracy. The Model-180 model showed an accuracy of 81.7%, while the other models showed an accuracy of less than 80%. Furthermore, the driving command prediction was not performed precisely in both the straight section and the turning section of the model, thereby resulting in low accuracy. Additionally, the accuracy of prediction at the boundary between the straight section and the turn section was low even in Model-360. However, Model-360 showed higher accuracy than other models for the left-wheel acceleration command and the right-wheel acceleration command when the vehicle body was biased in the straight section.

Conversely, Model-18 and Model-45 showed lower accuracy than Model-8, which had the smallest number of inputs. Furthermore, Model-8, Model-18, and Model-45 did not show a significant difference even in the F1 score of the congestion matrix (described in Section 5.1). These results indicate that a significant change in performance can occur only when the number of input data exceeds a certain threshold.

In this section, we evaluated the performance of the proposed neural networks. We designed a neural network model using only the range data measured from the LiDAR sensor. We divided it into six models according to the input size and compared the accuracy, learning time, latency, and battery consumption. As a result, the larger the input size, the higher the accuracy, but the latency increased. Battery consumption increased as the input size but did not increase significantly based on the total amount. Battery consumption remained the challenge of performance evaluation.

## 6. Conclusions and Future Work

In this study, resource consumption was measured when indoor autonomous driving was performed using neural networks; we evaluated the performance of the proposed neural network models. The proposed neural network learning receives the range data measured with the LiDAR sensor as an input and predicts a driving command suitable for the current situation. We conducted six neural network models according to the number of input data and performance evaluation to evaluate the effect of the number of input data on resource usage and performance. The confusion matrix, learning speed, response speed, battery consumption, and driving command accuracy were used as criteria for the performance evaluation.

Accuracy, precision, recall, and F1 score showed that performance improved as the number of input data increased in the confusion matrix. Furthermore, we compared the learning time required until the accuracy reached 0.95 in order to evaluate the learning speed. The accuracy of the model did not increase, or it took a long time to train, when the amount of input data was small; furthermore, the model with the most input data learned the fastest. The response time and battery consumption were affected by the number of input data. The number of input data increased as the response time increased by an average of 2%. The battery consumption increased by 0.55% on average. Model-360 showed the best performance, with an accuracy of 88.2% when predicting driving commands based on data measured during real-time driving. Furthermore, as regards accuracy, it was shown that a significant change in performance could be made only when the number of input data exceeded a certain level. However, the response time was delayed because of the large amount of input data, and battery usage increased. The delay in response had a negative effect on safety, such as on the possibility of collisions, and battery consumption affected the service time of an indoor autonomous vehicle. Therefore, it is necessary to determine a trade-off point by considering these conflicting characteristics during the design of a neural network model.

We designed the inference model using a single sensor and a single model. As a result, we did not observe a significant resource consumption gap. Additionally, dynamic vehicle models remain a challenge. In future work, we plan to conduct distributed learning using multiple vehicles and models. It will be possible to identify the distribution of resource consumption according to sensors and create a more improved vehicle model based on this.

## Figures and Tables

**Figure 1 sensors-23-02575-f001:**
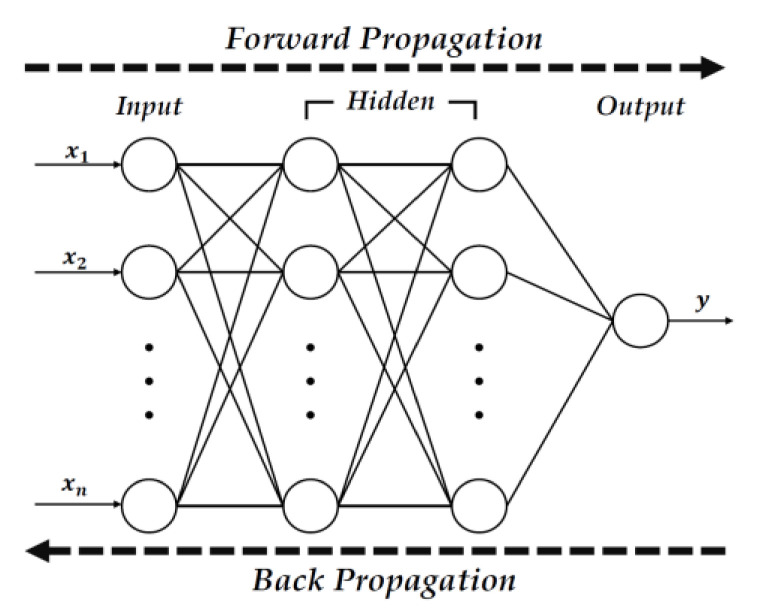
Structure of Back-propagation Neural Network.

**Figure 2 sensors-23-02575-f002:**
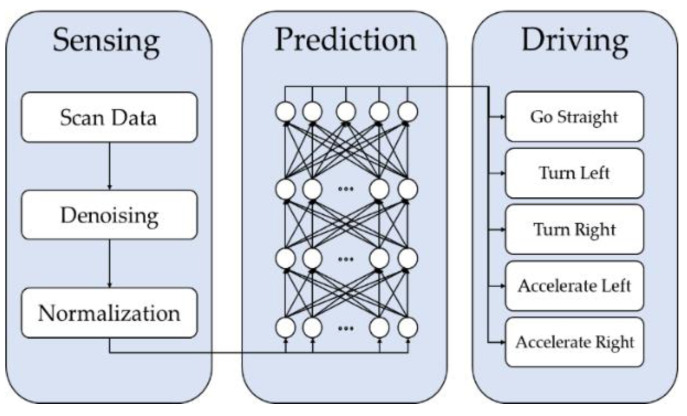
Indoor Autonomous Driving System with LiDAR.

**Figure 3 sensors-23-02575-f003:**
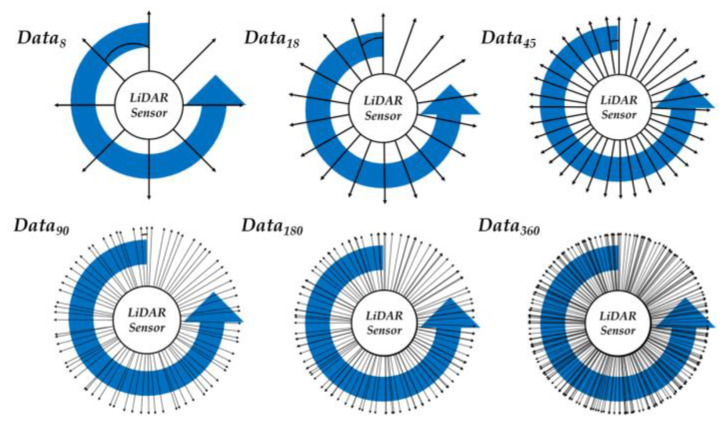
Data Density of Six Learning Models.

**Figure 4 sensors-23-02575-f004:**
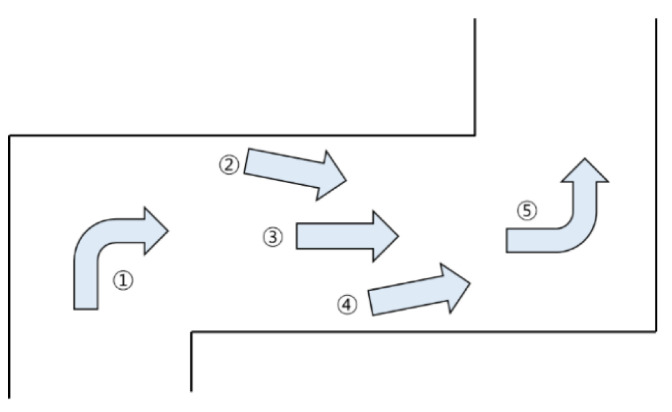
Autonomous Driving Commands.

**Figure 5 sensors-23-02575-f005:**
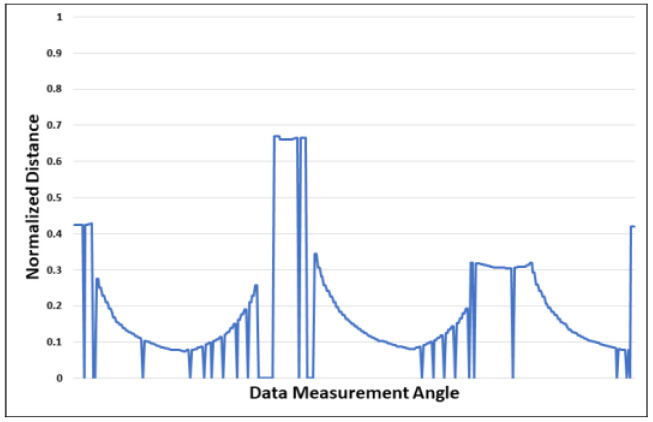
Normalized Range Data.

**Figure 6 sensors-23-02575-f006:**
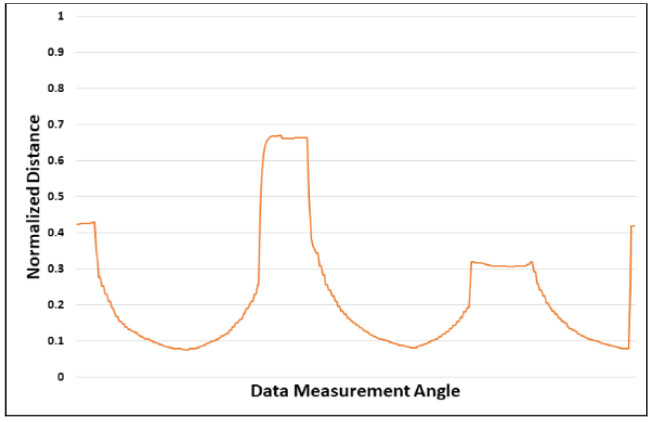
Normalized Range Data with Denoising.

**Figure 7 sensors-23-02575-f007:**
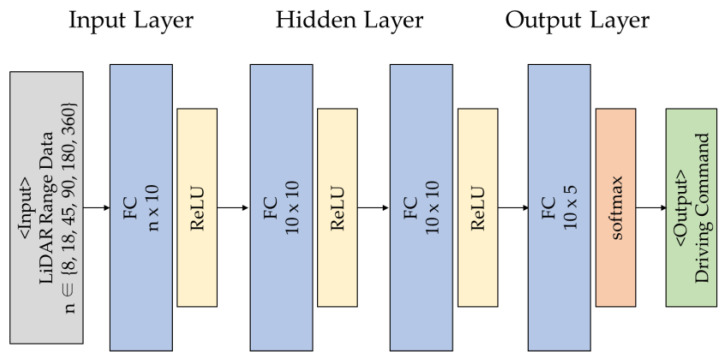
Neural Network Model for Indoor Autonomous Driving.

**Figure 8 sensors-23-02575-f008:**
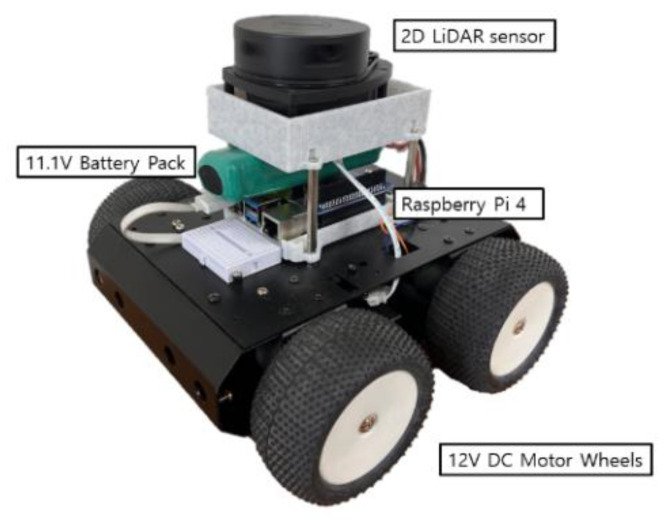
Autonomous Vehicle with LiDAR.

**Figure 9 sensors-23-02575-f009:**
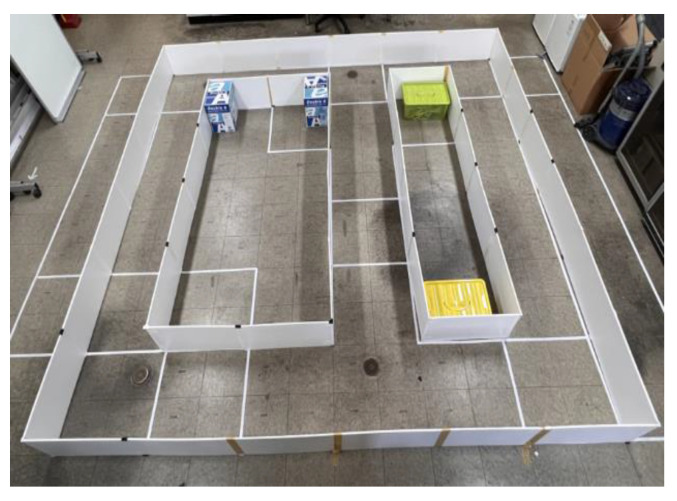
Indoor Autonomous Driving Track.

**Figure 10 sensors-23-02575-f010:**
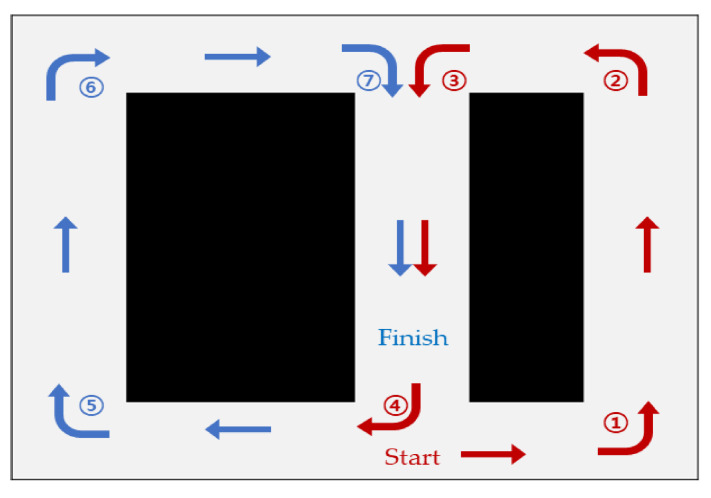
Path in a Driving Track.

**Figure 11 sensors-23-02575-f011:**
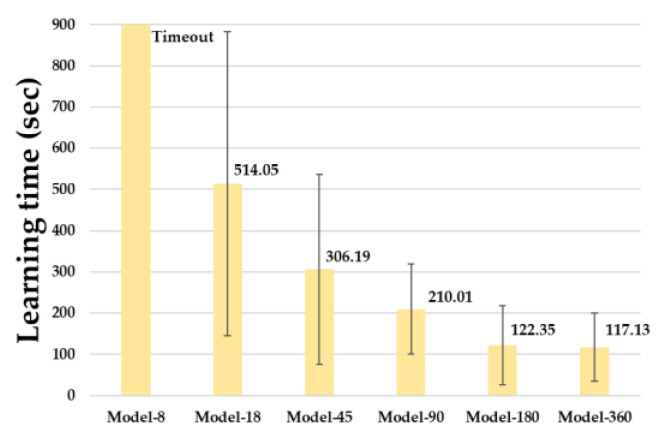
Learning Time of Prediction Models.

**Figure 12 sensors-23-02575-f012:**
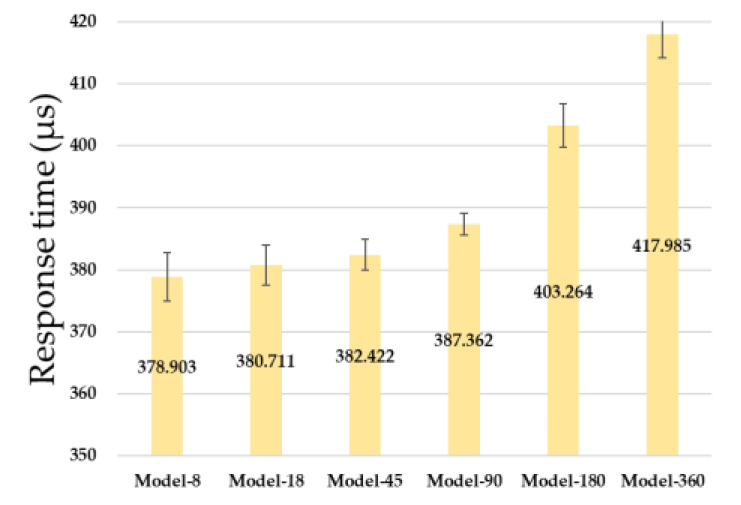
Response Time of six Neural Network Models.

**Figure 13 sensors-23-02575-f013:**
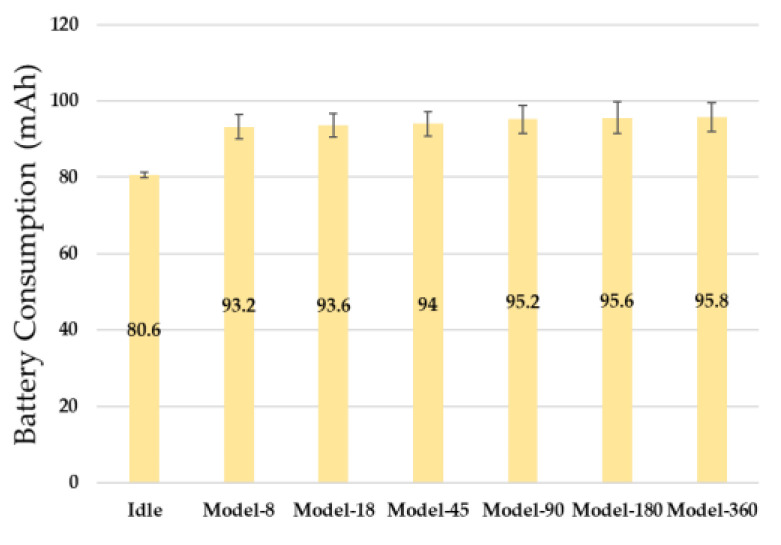
Battery Consumption of six Neural Network Models.

**Figure 14 sensors-23-02575-f014:**
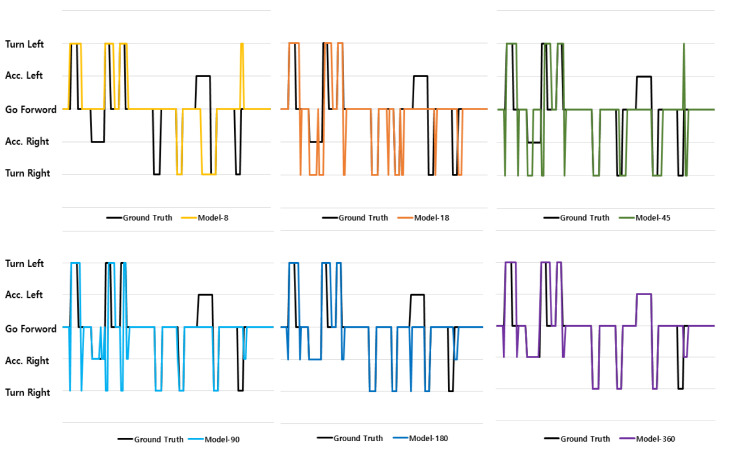
Driving Command Accuracy.

**Figure 15 sensors-23-02575-f015:**
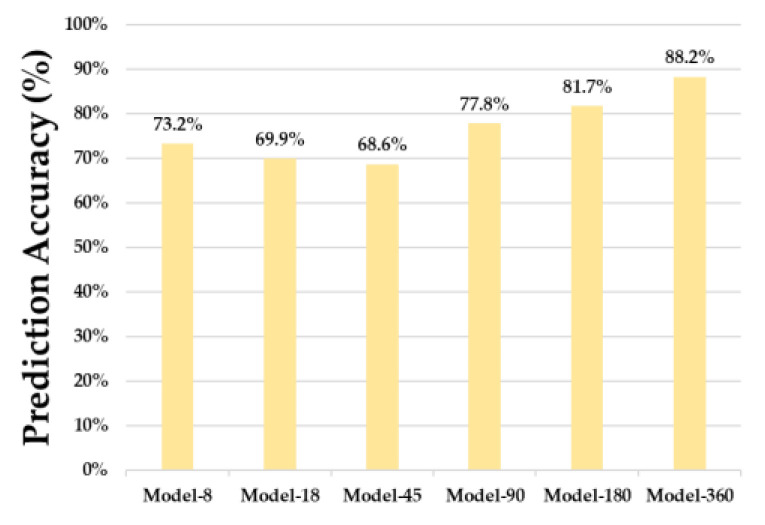
Prediction Accuracy while Driving.

**Table 1 sensors-23-02575-t001:** Specifications for Raspberry Pi and LiDAR.

	Raspberry Pi 4 Model B
CPU	Broadcom BCM2711, Cortex-A72 (ARM v8) 64-bit SoC @ 1.5 GHz
RAM	2GB LPDDR4-3200 SDRAM
OS	Ubuntu MATE 20.04 LTS
	RPLIDAR A1M8
Distance Range	0.15–12.0 m
Angular Range	0–360°
Scan Rate	5.5 Hz

**Table 2 sensors-23-02575-t002:** Part of Training Dataset.

R_0_	R_1_	...	R_358_	R_359_	C_0_	C_1_	C_2_	C_3_	C_4_
0.648	0.652	...	0.074	0.068	1	0	0	0	0
0.153	0.155	...	0.115	0.111	0	0	0	1	0
0.189	0.189	...	0.040	0.038	0	0	0	0	1
0.052	0.052	...	0.074	0.072	0	1	0	0	0
0.078	0.078	...	0.080	0.074	0	0	1	0	0

**Table 3 sensors-23-02575-t003:** Confusion Matrix for six Neural Network Models.

	Accuracy	Precision	Recall	F1 Score
Model-8	0.8953	0.4480	0.5504	0.4924
Model-18	0.9051	0.4705	0.5778	0.5157
Model-45	0.9101	0.4776	0.587	0.5237
Model-90	0.9269	0.5735	0.6676	0.6143
Model-180	0.9442	0.6784	0.7493	0.7086
Model-360	0.9661	0.8107	0.85	0.8275

## Data Availability

Not applicable.

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
