# Peer review of "Neural Network Models for Driving Control of Indoor Autonomous Vehicles in Mobile Edge Computing"

_sensors, 2023, doi:10.3390/s23052575_

Round 1

Reviewer 1 Report

The paper presents a neural network-based methodology to drive an autonomous vehicle using mobile edge computing. This work fits very well into the current research framework and the results presented are very interesting. Nevertheless, I recommend strengthening the current version of the paper by addressing the following points

- the authors should strengthen the rationale behind this work. Why is it necessary to have autonomous robots in an indoor environment? What are the limitations of the algorithms proposed in the literature for autonomous navigation that are overcome with the proposed approach?

- I suggest that the authors consider recent works on autonomous navigation to be cited in the state of the art. For example, mobile bases introduced for indoor use in the covid era are the following:

Tamantini, Christian, et al. "A robotic health-care assistant for COVID-19 emergency: A proposed solution for logistics and disinfection in a hospital environment." IEEE Robotics & Automation Magazine 28.1 (2021): 71-81.

Tiseni, Luca, et al. "UV-C mobile robots with optimized path planning: algorithm design and on-field measurements to improve surface disinfection against SARS-CoV-2." IEEE Robotics & Automation Magazine 28.1 (2021): 59-70.

- There are five commands assigned to the mobile base: go straight, turn left, turn right, accelerate left, and accelerate right. Why are other commands such as stop and reverse also not considered? The authors should add a discussion on this.

- The authors chose to tackle the classification problem using a neural network with a certain architecture. The authors should discuss what the advantages of an artificial neural network approach are compared to the use of traditional machine learning. Furthermore, the architecture of the network should be discussed further. 

- All results report average values obtained over 10 repetitions. In fact, the authors state: "to ensure the reliability of the performance...each model was trained 10 times". My question is: was cross-validation applied or did the training and validation dataset remain the same in all 10 trainings? The authors should comment further.

- In all graphs of the results, average values are shown. I suggest the authors also add the standard deviations to appreciate the distribution of the results.

- In the Performance Evaluation While Driving phase, how was ground truth obtained? How was real-time data labelled?

Author Response

We are indebted to the reviewers for their valuable comments, which were extremely helpful in revising our manuscript. We have given due consideration to each comment in preparing the revised manuscript.

Reviewer 2 Report

This paper proposes neural network models as a machine-learning method for autonomous driving in an indoor environment. This study uses as basis a LiDar sensor to generate information for neural network model in order to predict the most appropiate driving for the current location. In addition, it was used a Raspberry PI as main hardware for driving and learning way. It was implemented six neural netword models.

Performance evaluation results reveal a trade-off between the permanence and safety of indoor autonomous vehicles operating in an edge environment.

In these studies, the focus was on controlling the movement of autonomous vehicles rather than on map production and location recognition for vehicle movement.

The related works session should have been used to point out works at the frontier of knowledge involving lidar and edge computing, however, this did not happen. Space was used to explain how to use lidar among other aspects and few articles were related.

The autonomous driving vehicle uses the distance data collected from the LiDAR sensor to determine a suitable driving command for the current state of the vehicle and moves by itself.

In section 3.1, the set of equations for Lidar is not clear. It needs to be better detailed. Is it not clear what the main reason for using ReLU and SoftMax as activation functions at the middle level and command classification at the last level, respectively?

In a real vehicle model, the speed and the angle of rotation of the steering wheel are important parameters to allow the PID controllers to work satisfactorily. At no point do you talk about small-scale vehicle model or dynamic vehicle model. It would be very important to mention the limitations of your approach and how it could be used in practice in a real solution.

It is not clear what is running on cloud, edge and device. Likewise, the tradeoffs between them, limitations and infrastructure of each instance are not indicated.

In the path shown in figure 9, when an error occurs in the classifier, what happens? Can it crash because it hit the wall? Because there is no reverse command, right? What happens in these cases, for example?

Why did the work not apply other types of classifiers? This should be explained in the text. In addition, all initial neural network input data must be indicated.

You mention 3 steps in the sensing step but you don't detail all of them in the text. How was the controller implemented? What is the sampling of command submission data on RaspberryPI? A pseudocode would help in this understanding.

Author Response

(The authors gave the same response as above.)

Reviewer 3 Report

The authors propose in their study neural network models as a machine learning method for autonomous indoor driving, where the vehicle predicts the most appropriate driving based on the measurements made by the LiDAR sensor. The manuscript is well aligned with the content of the journal "Sensors" as well as the section "Vehicular Sensing".

Furthermore, the subject matter of the paper is of particular interest, and as a reviewer, I appreciate the authors' effort to illustrate it. Here are some key suggestions for authors to improve the quality of their work:

1. The sections: "Introduction and Related Work", allow space for a larger number of citations. The 25 bibliographical references selected by the authors are insufficient for a study of their claim in this area of research. A manuscript without this contribution makes it difficult for the reader to understand all the aspects that the authors wish to cover, as well as to detect the indicators of strengths and weaknesses of the proposed neural network models in a properly structured and justified way.

2. In relation to the previous point, the authors should compare in a more in-depth way their proposal with other models of activities implemented and proposed by other researchers for indoor autonomous driving. This part is key to be able to make a greater contrast of their findings and to include a greater reflection citing other research in the conclusions so that they are more solid in relation to the results obtained by their research.

3. Also, a discussion section must be well defined or within section “5. Performance Evaluation” where, in addition to mentioning the results obtained throughout the research, the authors highlight the contributions made in the field of research through a greater number of citations previously referred to in the previous sections of the manuscript.

4. In addition, the authors should indicate the limitations and gaps in their research, which may be resolved in the future or by other researchers in a correctly contrasted manner.

5. Finally, authors should elaborate on what findings have been obtained by other related research, what their most frequent limitations are, if there is a strong convergence between the theses with different methods, if there are elements of divergence, etc.

Author Response

(The authors gave the same response as above.)

Round 2

Reviewer 2 Report

Thank you for the noted reviews. I believe the article has been substantially revised. I believe that the answers are consistent in most of the questions raised.